

# Interaction-induced strong zero modes in short quantum dot chains with time-reversal symmetry

A. Mert Bozkurt[1*], Sebastian Miles[1], Sebastiaan L. D. ten Haaf[1], Chun-Xiao Liu[1], Fabian Hassler[2] and Michael Wimmer[1†]

**1** QuTech and Kavli Institute of Nanoscience, Delft University of Technology, P.O. Box 4056, 2600 GA Delft, The Netherlands
**2** Institute for Quantum Information, RWTH Aachen University, 52056 Aachen, Germany

⋆ a.mertbozkurt@gmail.com , † m.t.wimmer@tudelft.nl

## Abstract

We theoretically explore the emergence of strong zero modes in a two-site chain consisting of two quantum dots coupled due to a central dot that mediates electron hopping and singlet superconducting pairing. In the presence of time-reversal symmetry, the on-site Coulomb interaction leads to a three-fold ground-state degeneracy when tuning the system to a sweet spot as a function of the inter-dot couplings. This degeneracy is protected against changes of the dot energies in the same way as "poor man's" Majorana bound states in short Kitaev chains. In the limit of strong interactions, this protection is maximal and the entire spectrum becomes triply degenerate, indicating the emergence of a "poor man's" version of a strong zero mode. We explain the degeneracy and protection by constructing corresponding Majorana Kramers-pair operators and $\mathbb{Z}_3$-parafermion operators. The strong zero modes share many properties of Majorana bound states in short Kitaev chains, including the stability of zero-bias peaks in the conductance and the behavior upon coupling to an additional quantum dot. However, they can be distinguished through finite-bias spectroscopy and the exhibit a different behavior when scaling to longer chains.

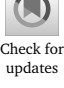

# 1 Introduction

Arrays of quantum dots offer a platform for quantum simulation of strongly-correlated and topological phases [1–4]. With a superconducting coupling in the form of crossed Andreev reflections, quantum dots have been proposed to implement the Kitaev chain which can be tuned into a topological phase [5]. Recently, it has been shown that both crossed-Andreev reflection (CAR) and elastic co-tunneling (ECT) between two quantum dots can be effectively tuned by an additional proximitized quantum dot between two normal quantum dots [6]. This has allowed to implement high-performance Cooper pair splitters [7–9] and to explore Majorana physics in a minimal Kitaev chain of two sites [10–12] and three sites [13]. When the quantum dots are in the spin-polarized regime and the amplitudes of these two processes are equal, a condition referred to as the *sweet spot*, a double quantum dot system connected by an ABS can feature Majorana bound states localized on the outer dots [6,14–16], so-called poor man's Majoranas (PMMs).

In these quantum dot systems, the charging energy $U$ is typically the largest energy scale. In current experiments, $U$ is of order several meV, whereas the inter-dot coupling is of order $30 - 80\mu eV$ [10–12]. This two-orders-of-magnitude difference in energy scales evokes the question of the role of interactions in these systems. The presence of strong charging energy makes the quantum dot platform fundamentally different from the original Majorana proposal in nanowires [17,18], and insights from those systems may not directly apply here. For instance, can interactions lead to false positives in the search for Majorana bound states in quantum dot systems? On the other hand, can interactions be used to engineer new types of states in these systems? The exploration of these two questions is the main goal of this study.

The importance of these questions is highlighted by a recent experimental work [11] implementing an artificial Kitaev chain with two sites in a proximitized two-dimensional electron gas. This experiment revealed stable zero-bias peaks for finite magnetic field, interpreted as PMMs. However, measurements also revealed a stable zero-bias peak in the *absence* of a magnetic field. In fact, the zero-bias conductance features were remarkably similar regardless of the value of magnetic field, despite PMMs only being expected at sufficiently large Zeeman splitting. This raises the question whether signatures of PMMs can be mimicked by trivial mechanisms in quantum dot systems. At the same time, a setup similar to the experiment was predicted theoretically to exhibit Majorana zero modes induced by Coulomb interaction in the presence of only a small Zeeman splitting [19]. Hence, it equally seems possible to induce precursors of topological states in quantum dot systems by interactions. Overall, this underlines the need for a systematic understanding of the zero-field case.

In this manuscript, we investigate strongly interacting double quantum dot system coupled by normal hopping and singlet superconducting inter-dot coupling via an additional proximitized quantum dot in the presence of time-reversal symmetry. We find that any finite charging energy on the quantum dots allows for a sweet spot characterized by a triply degenerate ground state. This ground state degeneracy is protected quadratically against changes of the on-site potential of either dot, akin to the two-site spinless Kitaev chain case. In the limit of large Coulomb interaction, the triple ground state degeneracy becomes completely protected against local changes of the on-site energies. We show that the system in this limit exhibits a poor man's version of strong zero modes, and construct corresponding Majorana Kramers-pair operators as well as $\mathbb{Z}_3$-parafermion operators explaining the protection against local perturbations. Moreover, just as in the spinless two-site Kitaev chain case [20], the ground state degeneracy is not lifted by coupling a third normal dot to the system via normal hopping. However, we can distinguish the zero-field, interaction-induced strong zero modes from PMMs through finite-bias spectroscopy and the absence of scaling to longer chains.

## 2 Charge stability diagram and transport properties of a double-quantum dot system

We consider a double-quantum dot system coupled by ECT and CAR processes, as sketched in Fig. 1(a). The Hamiltonian of this system is given by [19, 21]

$$H = \sum_{i,\sigma} \epsilon_i n_{i\sigma} + \sum_i U_i n_{i\uparrow} n_{i\downarrow} + t \sum_\sigma c_{L\sigma}^\dagger c_{R\sigma} + \Delta \sum_\sigma \eta_\sigma c_{L\sigma}^\dagger c_{R\bar{\sigma}}^\dagger + \text{H.c.}, \tag{1}$$

where $i = L, R$ denotes the site index, $n_{i\sigma} = c_{i\sigma}^\dagger c_{i\sigma}$ is the number operator on site $i$ with spin $\sigma$, $\epsilon_i$ is the on-site energy, $U_i$ is the Coulomb energy of dot $i$, $t$ is the normal hopping and $\Delta$ is the singlet type of superconducting pairing between left and right dot.[1] The term $\eta_\sigma = (-1)^\sigma$ encodes the singlet pairing and $\bar{\sigma} = -\sigma$ denotes the opposite spin $\sigma = \uparrow, \downarrow$. We note that as we consider a system with time-reversal symmetry, we can gauge-away the spin-orbit coupling by redefining the spin quantization axis on each dot, as detailed in Appendix F. Consequently, the presence of spin-orbit coupling, and consequently triplet superconducting pairing, is not necessary for our investigation. Tuning the relative strength of $t$ and $\Delta$ can for example be achieved through changing the energy of an ABS in a hybrid segment or proximitized quantum dot [6], as indicated in lighter color in Fig. 1(a). In the main text, we will exclusively use the effective model (1). However, using a model that includes the ABS gives comparable results, as shown in App. B.

---

[1]Here, we choose a gauge such that $t, \Delta \in \mathbb{R}$.

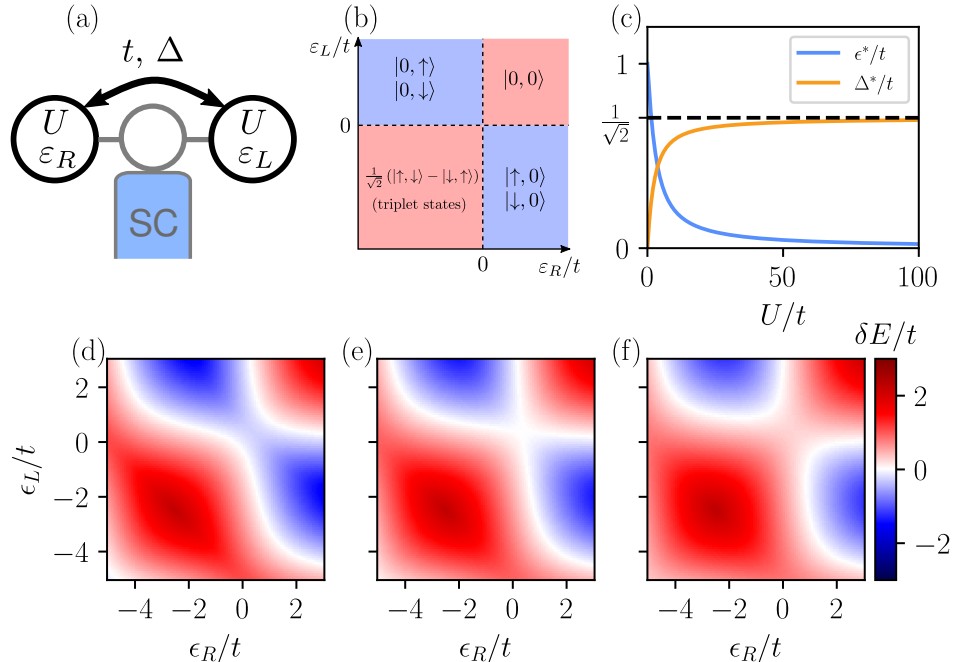

Figure 1: (a) Schematics of a two-site chain consisting of three quantum dots. The superconductor (blue) proximitizes the middle dot and facilitates normal hopping and superconducting pairing between the left and right quantum dots, each characterized by their respective on-site energies $\epsilon_i$ and charging energy $U$. (b) Sketch of the charge stability diagram, excluding the doubly-occupied states. The blue regions denote the odd fermion parity ground state, while the red regions represent the even fermion-parity ground state. (c) The parameters for the sweet spot, $\Delta^*$ and $\epsilon^*$, are depicted as a function of $U$. The evolution of the charge stability diagram for (d) $\Delta^* > \Delta = 0.293t$, (e) $\Delta = \Delta^* = 0.493t$, and (f) $\Delta^* < \Delta = 0.693t$. Here, we use $U = 5t$.

The charge stability diagram (CSD) of Eq. (1) of a double-quantum dot system coupled by ECT and CAR processes in the absence of a magnetic field has been studied in [21]. We show a sketch of the charge stability diagram in the absence of inter-dot interactions in Fig. 1(b), focusing on the energy range where each dot can be either empty or singly-occupied. Due to time-reversal symmetry, all states in the odd parity sector are doubly degenerate (blue parts of the CSD). When both dots are occupied by one electron, there are four degenerate states when the dots are decoupled: one singlet and three triplet states. However, in the presence of any finite $t$ or $\Delta$, it was shown [21] that the triplet states are higher in energy. Hence, for our purposes it is sufficient to only consider the singlet state and thus the even parity sector generally is singly degenerate. Ref. [21] further showed that for finite inter-dot coupling, either the odd or the even parity sectors merge, as we confirm in Figs. 1(d) and (f) by varying $\Delta/t$. This is due to either the ground state energy being lowered differently depending on the relative strength of $t$ and $\Delta$. However, since the CSD connectivity can be completely changed, *it is always possible* to find a relative strength of $\Delta/t$ such that there is a crossing, which we refer to as a sweet-spot, as shown in Fig. 1(e).

Fig. 1(d)–(f) shows the charge stability diagram in the form of $\delta E = E_{gs}^{odd} - E_{gs}^{even}$ being the energy difference of the ground states with opposite fermion parity. At the sweet-spot, and in general for the white lines in the CSD, the energies of the even and odd-parity ground states are equal, and the ground state triply degenerate. Moreover, the crossing corresponding to a

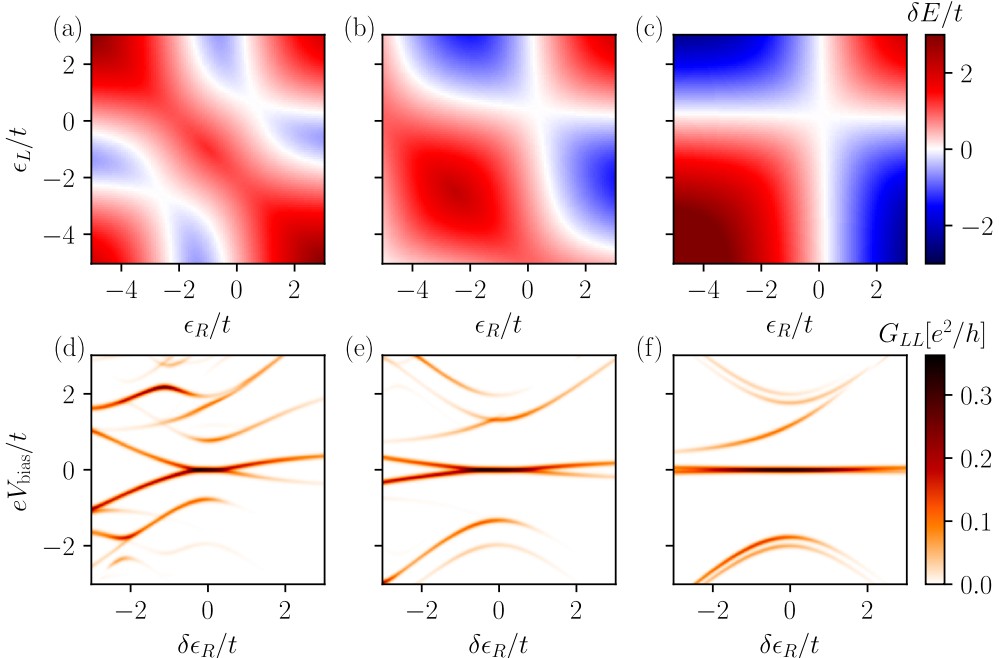

Figure 2: The charge stability diagram and local finite bias conductance as a function of the charging energy. Top panels: The charge stability diagram of the two-site spinful interacting chain for (a) $U = 2t$, (b) $U = 5t$, and (c) $U = 15t$, featuring a sweet spot. The crossing of the degeneracy lines at the sweet spot becomes straighter as $U$ increases. Additionally, as $U$ increases, the quadrants of the charge stability diagram move further away from each other. Bottom panels: The local finite bias conductance $G_{LL}$ for (d) $U = 2t$, (e) $U = 5t$, and (f) $U = 15t$, as a function of voltage bias $V_{\text{bias}}$ and the variation of the on-site energy on site $R$ away from the sweet spot, $\delta\epsilon_R$. The zero-bias peak persists for a wider range of detuning, $\delta\epsilon_R$, for larger local charging energy. Additionally, the local conductance feature visible in (f) for $\delta\epsilon_R < 0$ and $V_{\text{bias}} > 0$ describes the transport process via coupling of the ground state to the triplet states. For even larger voltage bias values, the conductance features exhibit splitting, which diminishes as $U$ increases. For transport simulations, we use dot-lead coupling $\Gamma = 0.0125t$ and reservoir temperature $T = 0.025t$.

sweet-spot represents a saddle point in $\delta E$. Hence, for small deviations around the sweet-spot, the three-fold ground state degeneracy is protected quadratically. This quadratic protection of ground state degeneracy is—up to the multiplicity of the degeneracy—identical to the spinless Kitaev chain case [14]. This is not surprising, as our arguments show that it is due to the intrinsic "topology" of the sweet spot, i.e. the fact the sweet spot must be a saddle point for $\delta E$. Hence we generally expect this quadratic protection when the connectivity of the CSD switches.

Note that this quadratic protection is seemingly in contradiction to Ref. [22] which claimed that the degeneracy in this system is changing linearly with changing the on-site energies $\epsilon_{R,L}$. This contradiction can be resolved by observing that Ref. [22] only considered degeneracies for $\epsilon_R = \epsilon_L = 0$. The sweet spot however is generally shifted away from zero on-site energy as shown in Fig. 1(c).

A hallmark of spinless PMMs is the persistence of the ground state degeneracy when changing only a single site on-site energy [14]. In general this does not apply to the degeneracies for the time-reversal symmetric Hamiltonian (1). In Fig. 2 we show the charge stability diagrams

(a)–(c) and the corresponding conductances, calculated using a rate-equation approach [15], for a normal probe on the left site (d)–(f) for different values of the Coulomb interaction $U$. The separation between the lower left quadrant of the charge stability diagram, comprised of states with double occupancy, from the upper right quadrant, comprised of empty dots, increases with Coulomb energy $U$. As a consequence, the degeneracy lines of the sweet spot crossing are initially tilted and become increasingly straighter with increasing $U$, Figs. 2(a)–(c). Hence, the ground state degeneracy becomes increasingly better protected against *local* potential changes, i.e. only changing $\epsilon_{R(L)}$ while keeping $\epsilon_{L(R)}$ at the sweet spot value. This can be directly observed in the behavior of the conductance that probes the excitation spectrum of the system. In particular, a ground state degeneracy gives rise to a zero-bias peak, whereas any splitting gives rise to a conductance only at finite bias.

As $U$ increases, the ground state degeneracy becomes more and more protected against the changes in one of the local on-site energies. This protection is reflected as a robust zero-bias peak in the local conductance spectroscopy on the left dot, as shown in Fig. 2(d)–(f). In fact, for large values of $U$, the zero-bias conductance of the two-site spinful interacting chain described by Eq. (1) becomes indistinguishable from the zero-bias conductance of a two-site spinless Kitaev chain hosting PMMs. Hence, these interaction-induced zero-energy states could be mistaken for PMMs. However, they can be distinguished by additional features at finite-bias, in particular the feature at positive bias voltage that approaches zero as the on-site energy of one site is decreased. It originates from the triplet states, and allows to distinguish this system from the spinless Kitaev chain, as discussed in detail in Appendix A.

## 3 Strong zero modes in the $U \to \infty$ limit

### 3.1 Eigenstates and eigenspectrum in the $U \to \infty$ limit

While it is possible to find a sweet spot where the ground state is triply degenerate for any finite charging energy $U$, the protection of the ground state degeneracy with respect to local changes is only truly possible in the limit of $U \to \infty$.[2] In this limit, double occupancy of a quantum dot is forbidden. This constraint can be implemented in Eq. (1) by replacing all fermionic operators by constrained fermions [23]. The constrained fermions are defined by the Hubbard operators $\bar{c}_{i\sigma} = (1 - n_{i\bar{\sigma}}) c_{i\sigma}$. The Hamiltonian then takes the form

$$H = \sum_i \epsilon_i \bar{n}_i + t \sum_\sigma \bar{c}_{L\sigma}^\dagger \bar{c}_{R\sigma} + \Delta \sum_\sigma \eta_\sigma \bar{c}_{L\sigma}^\dagger \bar{c}_{R\bar{\sigma}}^\dagger + \text{H.c.}, \tag{2}$$

where $\bar{n}_i = \sum_\sigma \bar{c}_{i\sigma}^\dagger \bar{c}_{i\sigma}$.

In this limit, the many-body energy levels for the odd parity sector are $\frac{1}{2}(\epsilon_L + \epsilon_R) \pm \left[t^2 + \frac{1}{4}(\epsilon_L - \epsilon_R)^2\right]^{1/2}$ with a multiplicity of 2 due to Kramers' degeneracy. For the even parity sector, the energy levels consist of $(\epsilon_L + \epsilon_R)$ with a multiplicity of 3, describing triplet states, and $\frac{1}{2}(\epsilon_L + \epsilon_R) \pm \left[2\Delta^2 + \frac{1}{4}(\epsilon_L + \epsilon_R)^2\right]^{1/2}$, describing singlet states. Therefore, when $t = \sqrt{2}\Delta$ and $\epsilon_L = \epsilon_R = 0$, the ground state becomes triply degenerate with an energy

---

[2]In current experimental implementations, $U$ exceeds all other energy scales in the system [10–12]. In this case, coupling to doubly occupied states is strongly suppressed and corrections through these states would only enter perturbatively and become visible through splittings in the excited states (see Fig. 2).

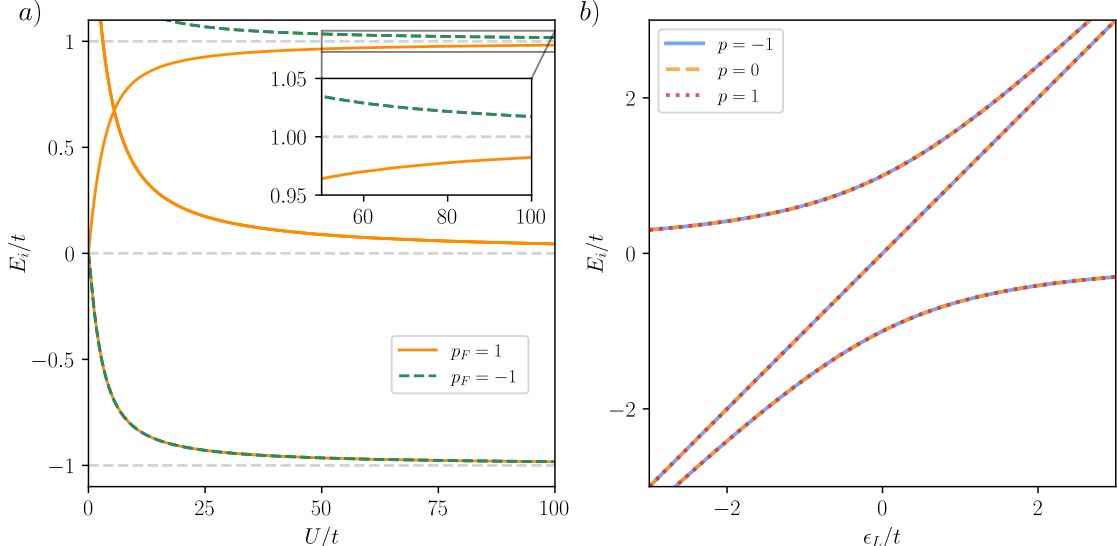

Figure 3: Many-body energy spectrum of the double-quantum dot system. In panel a), we demonstrate the many-body energy spectrum at the sweet spot as a function of $U$. The different colors indicate the different total fermion parity eigenvalues $p_F = \langle\psi|\sum_i(1-2n_i)|\psi\rangle$ of the corresponding eigenstate. The second excited state manifold becomes three-fold degenerate as $U \to \infty$. In addition, the states forming the first excited state manifold for $U \to \infty$ are three-fold degenerate for all $U$. Hence, the full many-body spectrum is three-fold degenerate for $U \to \infty$. Panel b) shows the many-body spectrum for $U \to \infty$ and protection of the three-fold degenerate structure in the many-body spectrum with respect to changes in $\epsilon_L$. The states with distinct generalized $\mathbb{Z}_3$ parity eigenvalues $p$ are differentiated by their colors and line styles.

of $E_g = -t$ [19]. The many-body eigenstates of the ground state manifold are

$$|n=0,\downarrow\rangle = \frac{1}{\sqrt{2}}\left(|0\downarrow\rangle - |\downarrow 0\rangle\right), \tag{3a}$$

$$|n=0,\uparrow\rangle = \frac{1}{\sqrt{2}}\left(|\uparrow 0\rangle - |0\uparrow\rangle\right), \tag{3b}$$

$$|n=0,S\rangle = -\frac{1}{\sqrt{2}}|00\rangle + \frac{1}{2}\left(|\uparrow\downarrow\rangle - |\downarrow\uparrow\rangle\right), \tag{3c}$$

where $n = 0$ denotes the ground state manifold (the states $n = 1, 2$ are given in Appendix C), we label odd-parity states with their spins $|\uparrow(\downarrow)\rangle$ and the even-parity ground state is a superposition of vacuum state and a singlet state $|S\rangle$. These eigenstates are reminiscent of the eigenstates of a two-site spinless Kitaev chain [14], except that the component with both dots occupied has a singlet character. As we show below, this leads to non-local correlations.

However, the three-fold degeneracy extends beyond the ground state manifold in this system. In fact, the many-body spectrum of the system consists of three different manifolds with three-fold degeneracy. As the entire many-body spectrum exhibits the three-fold degenerate structure, the zero energy excitations associated with this system are strong zero modes [24]. In Fig. 3(b), we show the many-body spectrum of the two-site chain as we vary one of the on-site energies $\epsilon_i$. The three-fold degeneracy of each three manifold is maintained upon varying one local on-site energy, demonstrating the protection of strong zero modes to this perturbation. Within these three manifolds, both the ground state ($n = 0$) and second excited

state ($n = 2$) manifolds each feature two odd and one even parity states. In contrast, the first excited state ($n = 1$) manifold comprises three triplet states with eigenvalues $E = 0$. We want to stress that the strong zero modes in our system only exist in the limit of $U \to \infty$. For any finite $U$, the entire many-body spectrum does not feature three-fold degenerate manifolds, as shown in Fig. 3 a), and hence, the resulting zero modes are weak zero modes.

## 3.2 Majorana Kramers-pair operators

Given the shared parity structure of the ground state and the second excited state manifolds, featuring two odd and one even fermion parity states, we introduce Majorana operators that allow switching between different parity states across the spectrum. Importantly, we exclude the first excited state manifold, as it does not permit any parity-switching zero-energy excitations and the triplet states do not couple to any of the other states by any term in the Hamiltonian. The modes described by these operators are deemed strong Majorana zero modes due to the consistent degeneracy throughout the spectrum [25, 26]. Additionally, recognizing that the odd states within each manifold are Kramers partners, we also associate Majorana operators with their Kramers partners. Based on these restrictions, Majorana Kramers-pair operators satisfy the conditions

$$\gamma_\sigma = \gamma_\sigma^\dagger, \tag{4a}$$

$$\gamma_\sigma |n, S\rangle = e^{i\phi} |n, \sigma\rangle, \tag{4b}$$

$$\gamma_\sigma |n, \sigma\rangle = e^{-i\phi} |n, S\rangle, \tag{4c}$$

where $\phi$ is a phase.

We then use the eigenstates of the many-body Hamiltonian and construct the Majorana Kramers-pair operators that satisfy Eq. (4) for a given spin projection.[3] We find these Majorana Kramers-pair operators as

$$\gamma_{R\sigma} = \eta_\sigma (1 - \bar{n}_L) \bar{c}_{R\sigma} - \frac{1}{\sqrt{2}} \left( \bar{n}_{L\sigma} \bar{c}_{R\bar{\sigma}} - \bar{c}_{L\sigma}^\dagger \bar{c}_{L\bar{\sigma}} \bar{c}_{R\sigma} \right) + \text{H.c.}, \tag{5a}$$

$$\gamma_{L\sigma} = i\eta_\sigma (1 - \bar{n}_R) \bar{c}_{L\sigma} + \frac{i}{\sqrt{2}} \left( \bar{n}_{R\sigma} \bar{c}_{L\bar{\sigma}} - \bar{c}_{R\sigma}^\dagger \bar{c}_{R\bar{\sigma}} \bar{c}_{L\sigma} \right) + \text{H.c.}, \tag{5b}$$

where $\eta_\sigma = (-1)^\sigma$ and $\bar{\sigma} = -\sigma$ denotes the opposite spin. The strong correlation in the system is evident from the presence of products of number operators in the definition of Majorana operators, and by products of operators flipping the spin on a dot. The latter are related to the fact that the even ground state involves a spin singlet state.

The Majorana Kramers-pairs operators given in Eq. (5) commute with the Hamiltonian at the sweet spot by construction. Furthermore, each Majorana Kramers-pair operator commutes with one of the number operators $\bar{n}_i$, specifically $[\bar{n}_L, \gamma_{R\sigma}] = [\bar{n}_R, \gamma_{L,\sigma}] = 0$. This explains why any perturbation involving only one of the on-site energies $\epsilon_i$ will not lift the degeneracies within the $n = 0, 2$ states. We note that in terms of the commutation relations with the number operators on each dot, these Majorana Kramers-pairs operators are local. However, in terms of dot creation and annihilation operators, they clearly are not.

It is worth emphasizing that these Majorana Kramers-pairs, or in other words the corresponding ground state degeneracy, only exist due to interactions: A no-go theorem states that Majorana Kramers-pairs cannot be realized in non-interacting electronic systems with a single conventional superconductor [27]. Hence, the charging energy $U$ is the driving force for obtaining the ground state degeneracy.

---

[3]We refer the reader to Appendix D for more details on how to construct the Majorana operators from the eigenstates.

### 3.3 $\mathbb{Z}_3$-parafermion operators

Majorana Kramers-pairs operators can only be meaningfully defined in terms of the manifolds $n = 0, 2$ containing even and odd parity states. In the following, we will introduce a different, complementary description that takes the full spectrum into account.

Having a many-body spectrum that is three-fold degenerate signals a symmetry of the system. Beyond the fermion-parity conservation, the system has the additional generalized-parity symmetry

$$P_{\mathbb{Z}_3} = \omega^{\sum_j (n_{j\uparrow} + 2n_{j\downarrow})}, \tag{6}$$

with $\omega = e^{i2\pi/3}$ [28] and $n_{j\sigma} = c_{j\sigma}^\dagger c_{j\sigma}$ the spin-resolved number operator defined on site $j$. We find that the eigenstates within each degenerate manifold $n$ are uniquely characterized by their corresponding generalized parity eigenvalue $p = 0, 1, 2$

$$P_{\mathbb{Z}_3} |n, p\rangle = \omega^p |n, p\rangle . \tag{7}$$

As all the states $|n, p\rangle$ for fixed $n$, are degenerate, we can construct a parafermion operator $\chi$. These operators switch between eigenstates with different $P_{\mathbb{Z}_3}$-parity eigenvalues $p$ within each degenerate manifold with

$$\chi |n, p\rangle = a_{n,p} |n, p + 1 \,(\text{mod } 3)\rangle , \tag{8a}$$

$$\chi^3 = \mathbb{1} , \tag{8b}$$

$$\chi P_{\mathbb{Z}_3} = \omega P_{\mathbb{Z}_3} \chi , \tag{8c}$$

where the coefficients $a_{n,p}$ are complex and satisfy $\prod_p a_{n,p} = 1$ for all $n$, ensuring that $\chi^3 = \mathbb{1}$. Note that the parafermion operators do not obey superselection as they must contain both fermion-parity switching and conserving operators.

To construct the parafermion operators, we use the many-body eigenstates of the system. In addition to satisfying the conditions outlined for parafermion operators in Eq. (8), we require that these operators commute with one of the number operators. This requirement helps explain how the many-body spectrum is protected against changes in local on-site energies.[4] We find two parafermion operators $\chi_L$ and $\chi_R$ expressed in terms of constrained fermion operators as

$$\chi_R = (1 - \bar{n}_L)\left(-\bar{c}_{R\downarrow}^\dagger + \bar{c}_{R\uparrow}\right) + \left(\bar{c}_{R\downarrow}^\dagger \bar{c}_{R\uparrow} + \frac{1}{\sqrt{2}}\left(\bar{c}_{R\downarrow} + \bar{c}_{R\uparrow}^\dagger\right)\right)\bar{c}_{L\downarrow}^\dagger \bar{c}_{L\uparrow}$$
$$- \frac{1}{\sqrt{2}}\left(\bar{n}_{L\uparrow}\bar{c}_{R\downarrow}^\dagger + \bar{n}_{L\downarrow}\bar{c}_{R\uparrow}\right) - \left(1 - \frac{1 + \sqrt{2}}{\sqrt{2}}\bar{n}_L\right)\bar{c}_{R\uparrow}^\dagger \bar{c}_{R\downarrow} - \left(1 - \frac{1 + \sqrt{2}}{\sqrt{2}}\bar{n}_R\right)\bar{c}_{L\uparrow}^\dagger \bar{c}_{L\downarrow}, \tag{9a}$$

$$\chi_L = (1 - \bar{n}_R)\left(\bar{c}_{L\downarrow}^\dagger + \bar{c}_{L\uparrow}\right) + \left(\bar{c}_{L\downarrow}^\dagger \bar{c}_{L\uparrow} + \frac{1}{\sqrt{2}}\left(\bar{c}_{L\uparrow}^\dagger - \bar{c}_{L\downarrow}\right)\right)\bar{c}_{R\downarrow}^\dagger \bar{c}_{R\uparrow}$$
$$- \frac{1}{\sqrt{2}}\left(\bar{n}_{R\uparrow}\bar{c}_{L\downarrow}^\dagger - \bar{n}_{R\downarrow}\bar{c}_{L\uparrow}\right) + \left(1 + \frac{1 - \sqrt{2}}{\sqrt{2}}\bar{n}_L\right)\bar{c}_{R\uparrow}^\dagger \bar{c}_{R\downarrow} + \left(1 + \frac{1 - \sqrt{2}}{\sqrt{2}}\bar{n}_R\right)\bar{c}_{L\uparrow}^\dagger \bar{c}_{L\downarrow}. \tag{9b}$$

The parafermion operators given in Eq. (9) commute with the Hamiltonian at the sweet spot and satisfy $\chi_i^3 = \mathbb{1}$ by construction. Furthermore, each parafermion operator commutes with one of the number operators $\bar{n}_i$, specifically $[\bar{n}_L, \chi_R] = [\bar{n}_R, \chi_L] = 0$. Hence these parafermion operators also explain the protection of the degeneracy in the many-body spectrum against variations in the on-site energies.

The parafermion operators given in Eq. (9) do not satisfy $\mathbb{Z}_3$ parastatistics. The reason for this is that we want the parafermion operators to commute with the number operators

---

[4]We refer the reader to Appendix E for the details.

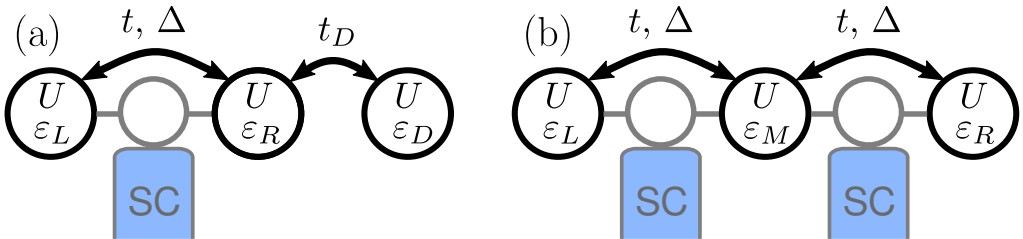

Figure 4: (a) Quantum dot test: A two-site spinful interacting chain is coupled to a quantum dot $D$ via a spin-conserving normal hopping. (b) Three-site spinful interacting chain with sites $L$, $M$ and $R$, each coupled via normal hopping $t$ and superconducting pairing $\Delta$.

to explain the robustness with respect to changes in on-site energies. If we remove this restriction, we can find coefficients $a_{n,p}$ such that the resulting parafermion operators obey the $\mathbb{Z}_3$ parastatistics $\chi_L \chi_R = \omega \chi_R \chi_L$.[5] As a interesting side-remark, we note that we were able to find parafermion operators that commute with the number operators and satisfy the $\mathbb{Z}_3$ parastatistics *when* projected to the ground state manifold.[6]

### 3.4 Low-energy effective Hamiltonian with parafermion operators

Next, we explore the low-energy physics of the three-fold degenerate ground state manifold. To this end, we project the parafermion operators given in Eq. (9) to the ground state manifold. These projected operators, denoted as $\tilde{\chi}_R$ and $\tilde{\chi}_L$, still commute with their respective number operators.

Mapping the parafermion operators onto the ground state allows us to derive the low-energy effective Hamiltonian

$$\tilde{H} = -\frac{\left(t + \sqrt{2}\Delta\right)}{2} \mathbb{1} + \frac{\left(t - \sqrt{2}\Delta\right)}{4} \left(\tilde{\chi}_L^\dagger \tilde{\chi}_R + \tilde{\chi}_R^\dagger \tilde{\chi}_L\right). \tag{10}$$

Note that the low-energy Hamiltonian is akin to the low-energy Hamiltonian of a two-site $\mathbb{Z}_3$ parafermion chain [24, 29–31]. The first term is an energy offset such that at the sweet spot, with $t = \sqrt{2}\Delta$, the ground state energy is $E_g = -t$.

## 4 Three-site spinful interacting chains

### 4.1 Quantum dot test

Having established the characterization of the two-site spinful interacting chain and its protection due to $P_{\mathbb{Z}_3}$ parity, we now investigate its behavior when the chain length is increased. To this end, we first consider adding a third spinful quantum dot only coupled by a normal hopping $t$, as show in Fig. 4(a). This system is the time-reversal symmetric variant of a quantum dot test originally designed for Majorana bound state detection. This test, aimed at identifying unpaired localized Majorana bound states, has been previously considered in various setups, including proximitized nanowires [32–34] and artificial Kitaev chains [20,35]. Here, we probe the two-site chain by using a test quantum dot $D$ in the single electron limit, i.e. $U_D \to \infty$,

---

[5]The choice of coefficients with $a_{n,p}^1 = \omega$ and $a_{n,p}^2 = \omega^{-p}$ yields two parafermion operators that obey $\mathbb{Z}_3$ parastatistics. In this case, the first parafermion operator still commutes with $n_R$, however the second parafermion operator does not commute with $n_L$.

[6]See Appendix E.2 for details.

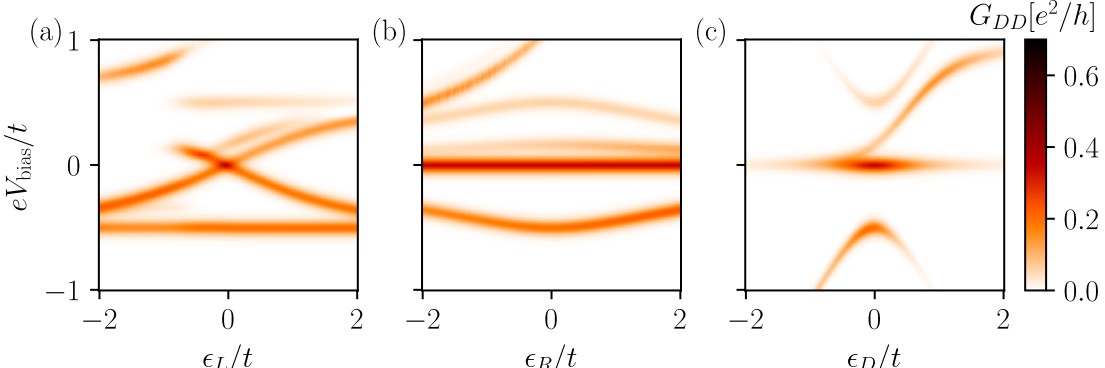

Figure 5: Quantum dot test: Another quantum dot $D$ is attached, coupled exclusively to site $R$ through spin-conserving hopping. Variation of (a) on-site energy $\epsilon_L$ results in a splitting of the zero-bias peak in the local finite bias conductance measured from quantum dot $D$. Conversely, changes in (b) on-site energy $\epsilon_R$ or (c) the test quantum dot $\epsilon_D$ do not induce a splitting of the zero-bias peak. For transport simulations, we use dot-lead coupling $\Gamma = 0.0125t$ and reservoir temperature $T = 0.025t$.

with an on-site energy $\epsilon_D = 0$. Quantum dot $D$ is coupled to site $R$ of the two-site chain with spin-conserving hopping $H_{RD} = t_D \sum_\sigma \bar{c}_{R\sigma}^\dagger \bar{c}_{D\sigma} + \text{H.c.}$, as shown in Fig. 4(a). We then measure local finite bias conductance $G_{DD}$ as we vary on-site energies of each of the three sites in the system as shown in Fig. 5.

In Fig. 5(a), we observe that detuning the on-site energy of site $L$ leads to a splitting in the zero-bias peak in the local differential conductance $G_{DD}$ measured by tunnel coupling a normal lead to quantum dot $D$. In contrast to varying the on-site energy of site $L$, varying the on-site energy of site $R$ or the test dot $D$ does not lead a splitting in the zero-bias conductance peak, as shown in Fig 5(b)–(c).

The outcome of the quantum dot test closely resembles the quantum dot test for poor man's Majorana zero modes [20]. There the splitting of the zero-bias peak, when the on-site energy of site $L$ is detuned, is attributed to the leakage of the left Majorana wavefunction to the right site. Then, the right site no longer hosts an isolated Majorana wavefunction and the zero-bias peak splits linearly. On the other hand, detuning the on-site energies of site $R$ or the test quantum dot would not lead to any splitting as there would be a single Majorana residing on the site $R$. Disregarding the interacting nature of our system, the outcomes of the quantum dot test could thus be (mis)interpreted as the presence of an isolated zero-mode in each dot.

To understand the role of the quantum dot test in our spinful interacting system, we construct a low-energy Hamiltonian using the $\mathbb{Z}_3$-parafermion operators that we constructed before to show the stability against changes in local potentials. To that end, we project the spin-conserving coupling term between site $R$ and test quantum dot $H_t = t_D \sum_\sigma \bar{c}_{R\sigma}^\dagger \bar{c}_{D\sigma} + \text{H.c.}$ to the ground state manifold. Then, the projected coupling Hamiltonian takes the form

$$\tilde{H}_t = \frac{t}{\sqrt{6}} \left( A_1 d_{D\uparrow}^\dagger + A_2 d_{D\downarrow}^\dagger + \text{H.c.} \right), \tag{11}$$

where operators $A_{1,2}$ act on site $L$ and site $R$ of the original two-site chain and are expressed in terms of parafermion operators

$$A_1 = \frac{1}{2}(\tilde{\chi}_L + \tilde{\chi}_R), \tag{12a}$$

$$A_2 = \frac{1}{2}(\tilde{\chi}_L^\dagger + \tilde{\chi}_L \tilde{\chi}_R), \tag{12b}$$

and operators

$$d^{\dagger}_{D,\sigma} = \sqrt{\frac{2}{3}}\left(\frac{1}{\sqrt{2}}\bar{c}^{\dagger}_{D\sigma} + \bar{c}_{D\bar{\sigma}}\right),\tag{13}$$

act on the test quantum dot states.

The form of Eq. (11) together with Eq. (12) indicate that the fermionic states in the quantum dot $D$ actually couple to both of the parafermions. Therefore, the result of the quantum dot test for our system cannot be interpreted as selectively coupling to a single parafermion, in contrast to Majorana bound states [20, 32, 33].

This leaves the question of why the quantum dot test leaves the ground state degeneracy unchanged. In fact, we find that the entire many-body spectrum of the combined three-dot system is also comprised by degenerate manifolds. The fact that the system still features $P_{\mathbb{Z}_3}$ symmetry, each degenerate manifold has eigenstates with three different generalized parity eigenvalues. This property allows us to construct two parafermion operators $\chi_1$ and $\chi_2$, similar to how we constructed parafermion operators for the two-site chain case given in Eq. (9).[7]

Each parafermion operator, in addition to commuting with the Hamiltonian at the sweet spot and $\bar{n}_D$, also commutes with either $\bar{n}_L$ or $\bar{n}_R$, specifically $[\chi_1, \bar{n}_L] = [\chi_2, \bar{n}_R] = 0$. On the other hand, only $\chi_2$ commutes with $H_{RD}$, the operator that describes spin-conserving hopping between site $R$ and quantum dot $D$. As a consequence, varying $\epsilon_L$ results in the splitting of the degenerate energy levels, whereas varying $\epsilon_R$ or $\epsilon_D$ does not.

A natural question to ask is whether parafermion operators in three-site and two-site cases are related. Given the strongly-correlated nature of the system, the form of these operators are quite involved, and involve terms mixing operators from all three dots. Nevertheless, we can project the parafermion operators for the three-site system onto a two-site system by tracing out the degrees of freedom related to quantum dot $D$. In this case, we recover that the projected three-site parafermion operators are identical to the parafermion operators for the two-site case

$$\text{Tr}_D \chi_1 = \chi_R,\tag{14a}$$

$$\text{Tr}_D \chi_2 = \chi_L.\tag{14b}$$

This equivalence underlines the protection mechanism for the degeneracies in these two setups as parafermion operators.

## 4.2 Absence of scaling

The presence of strong zero modes in a two-site system raises a key question: Can extending the chain to more sites bring about topologically protected zero modes? An example is seen in Majorana zero modes within an N-site Kitaev chain with uniform $t = \Delta$ for all hoppings and $\epsilon = 0$ for all on-site energies. To explore the emergence of such modes in a strongly interacting chain with time-reversal symmetry, we examine a three-site chain, with sites $L$, $M$ and $R$ as shown in Fig. 4(b), with normal hopping and superconducting pairing between adjacent sites induced by proximitized quantum dots. Given our focus on strong zero modes, we assume infinite charging energy in each site and use constrained fermion operators as detailed in Sec. 3.

---

[7]See App. E.3 for the construction of the parafermion operators.

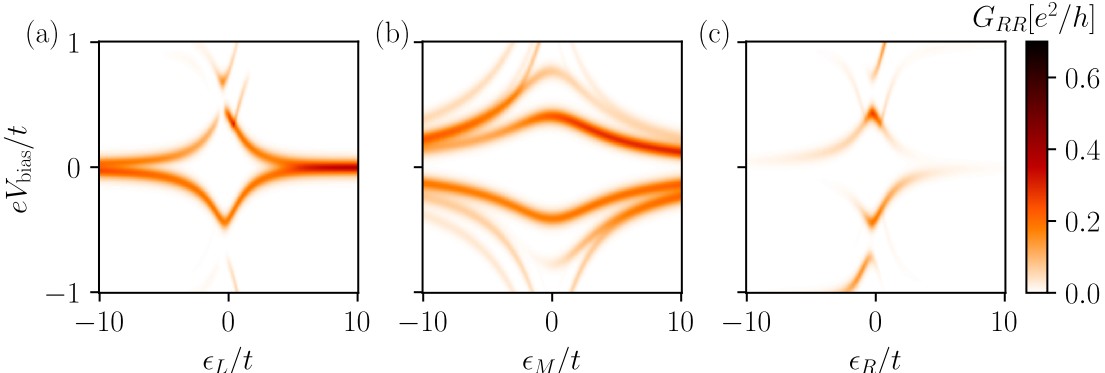

Figure 6: The local finite bias conductance for the three site chain shown in Fig. 4(b) as we detune (a) $\epsilon_L$, (b) $\epsilon_M$, and (c) $\epsilon_R$ on-site energies. At zero-detuning for each case, the system shows a gap, indicating the absence of ground state degeneracy with opposite fermion parity. For transport simulations, we use dot-lead coupling $\Gamma = 0.0125t$ and reservoir temperature $T = 0.025t$.

To investigate this, we set the condition $t = \sqrt{2}\Delta$ for all hopping magnitudes and $\epsilon = 0$ for all on-site energies. We find that the many-body ground state no longer maintains the triply degenerate structure with one even- and two odd-parity eigenstates. Instead, the ground state exhibits even fermion parity, accompanied by an excitation gap to the lowest odd fermion parity eigenstates. This aspect becomes apparent in the local finite bias conductance spectroscopy of the three-site chain as illustrated in Fig. 6. The absence of a zero-bias peak in Fig. 6, which signifies degenerate ground states with opposite fermion parities, is replaced by a gap in the excitation spectrum. As either $\epsilon_L$ or $\epsilon_R$ is detuned such that the site in question is depleted, the system effectively reduces again to a two-site chain. We observe this feature in local differential conductance shown in Fig. 6(a),(c) as a development of zero bias peak for $\epsilon_i \gg t$.

Hence, despite the zero-bias conductance being identical for a two-site spinless Kitaev chain and a spinful interacting two-site chain and despite the similarity for the quantum dot test in both cases, the spinful interacting three-site chain differs crucially from the spinless three-site Kitaev chain. We believe that this should be testable in current experiments.

## 5  Discussion and conclusion

In this work, we have studied spinful interacting quantum dots coupled by normal hoppings and singlet-type of superconducting pairings under time-reversal symmetry. The combination of local Coulomb interactions, normal hopping and singlet-type superconducting pairing within a two-site system results in a three-fold degenerate ground state, which is quadratically protected against changes in the on-site energies. This yields experimental features similar to regular Majorana zero modes in a two-site chain, although they can be distinguished through finite bias conductance spectroscopy. Hence, our results show that the presence of a sweet spot alone does not guarantee the existence of localized Majorana bound states.

In the limit of $U \rightarrow \infty$, the entire many-body spectrum features three-fold degenerate manifolds, revealing the emergence of strong zero modes. We find two different interpretations for the existence of such strong zero modes, namely Majorana Kramers-pairs and $\mathbb{Z}_3$ parafermions. We explicitly construct corresponding Majorana Kramer-pairs operators and $\mathbb{Z}_3$ parafermion operators. In particular, from the parafermion operators, we can understand

the protection of the degeneracy in the entire spectrum with respect to changes in the on-site energies and the coupling strength to the test quantum dot as discussed in Sec. 4.1. Projecting the parafermion operators of the two-site spinful interacting chain onto the ground state manifold yields a low-energy Hamiltonian, represented by Eq. (10), which resembles a two-site parafermion chain Hamiltonian. Moreover, by selecting appropriate phases for the parafermion operators, the projected operators obey $\mathbb{Z}_3$ parastatistics.

We find that these strong zero modes present in the two-site spinful interacting chain feature the same resilience as regular Majorana zero modes [20, 32, 33] against the quantum dot test. In contrast, however, extending the chain to more sites does not retain its triply-degenerate many-body spectrum. The deviation from the triply degenerate structure in the many-body spectrum for longer chains emphasizes the need for further investigation.

Previous studies [28, 36] have used Fock parafermions proposed in Ref. [37] to embed a parafermionic chain [29] in a fermionic system, resulting in fermionic Hamiltonians with parity breaking terms or three-body interaction terms that are hard to implement in experimental settings. Here, we start from a setup that can be realized experimentally [11] *and* construct parafermion operators for this system. Given that the system has two sites only, we call these modes "poor man's $\mathbb{Z}_3$ parafermions" in analogy to poor man's Majoranas [14]. However, there are several open questions: Can these strong zero modes obtain topological protection once extended to longer chains? Can we use this minimal model to demonstrate braiding or fusion for $\mathbb{Z}_3$ parafermions that could be used for universal quantum computation? Answering these questions may open promising avenues in strongly-correlated time-reversal invariant systems.

# Acknowledgments

We acknowledge useful discussions with Martin Leijnse, Michele Burello, Anton Akhmerov, Dirk Schuricht, Natalia Chepiga, Bowy La Rivière, Rubén Seoane Souto, Viktor Svensson, William Samuelson.

**Data availability**    The code used to generate the figures is available on Zenodo [38].

**Funding information**    This work was supported by funding from the Dutch Organization for Scientific Research (NWO) through OCENW.GROOT.2019.004, by a subsidy for top consortia for knowledge and innovation (TKI toeslag), and by funding from Microsoft Research. F.H. further acknowledges funding by the Deutsche Forschungsgemeinschaft (DFG, German Research Foundation) under Germany's Excellence Strategy – Cluster of Excellence Matter and Light for Quantum Computing (ML4Q) EXC 2004/1 – 390534769.

**Author contributions**    A.M.B. and M.W. defined the project scope. A.M.B., S.M. and M.W. developed the code. A.M.B. and S.M. performed the numerical simulations that generated the figures. S.L.D.t.H. and C.-X.L. performed numerical simulations in the early stages of the project. A.M.B., S.M., and F.H. performed the analytical calculations. A.M.B. prepared the figures. A.M.B., S.M., and M.W. wrote the manuscript with input from S.L.D.t.H., C.-X.L., and F.H. All authors analyzed the results. M.W. oversaw the project.

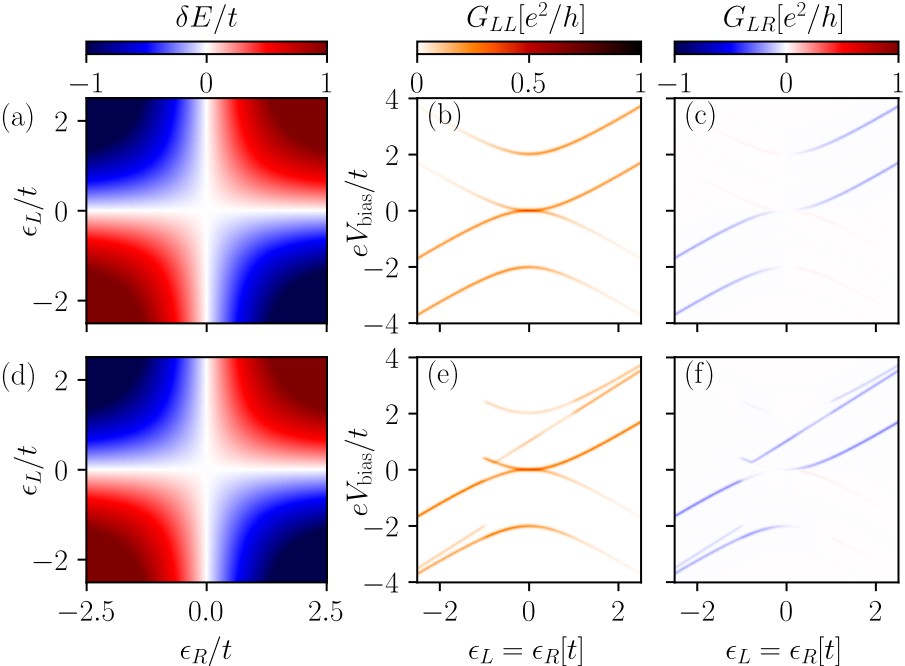

Figure 7: Comparison between a two-site spinless Kitaev chain and a two-site spinful interacting chain. Top panels: (a) depicts the charge stability diagram of the two-site Kitaev chain at the sweet spot. (b) illustrates the local and (c) nonlocal differential conductance of the two-site Kitaev chain at the sweet spot, plotted as a function of the voltage bias and simultaneous detuning of both on-site energies. Bottom panels: (d) depicts the charge stability diagram of the two-site spinful interacting chain in the limit $U \to \infty$. (e) shows the local and (f) nonlocal differential conductance of the two-site spinful interacting chain, plotted as a function the voltage bias and simultaneous detuning of both on-site energies. For transport simulations, we use dot-lead coupling $\Gamma = 0.0125t$ and reservoir temperature $T = 0.025t$.

# A  Experimental features of spinless Kitaev chain vs. spinful interacting chain

This section compares two-site spinless Kitaev chain to two-site spinful interacting chain, focusing on their charge stability diagrams and the resulting finite bias conductance spectroscopy. The many-body Hamiltonian for spinless two-site Kitaev chain is expressed as

$$H_{\text{Kitaev}} = \sum_{i=L,R} \epsilon_i c_i^\dagger c_i + t c_L^\dagger c_R + \Delta c_L^\dagger c_R^\dagger + \text{H.c.} \tag{A.1}$$

The sweet spot condition for two-site spinless Kitaev chain requires $\epsilon_i = 0$ and $t = \Delta$, leading to a two-fold degenerate many-body spectrum. This degeneracy becomes apparent in the charge stability diagram illustrated in Fig. 7(a), where detuning the on-site energies causes the degeneracies to split. The impact is also reflected in the local finite bias spectroscopy, depicted in Fig. 7(b), where the zero-bias peak splits upon detuning both on-site energies by $\epsilon \equiv \epsilon_L = \epsilon_R$. For completeness, in Fig. 7(c), we show the nonlocal finite bias conductance $G_{LR}$ as both on-site energies are varied. In comparison with the poor man's Majorana zero modes, we illustrate the charge stability diagram and finite bias conductance spectroscopy for two-site spinful interacting chain in Fig. 7(d-f). Although the charge stability diagrams for each system

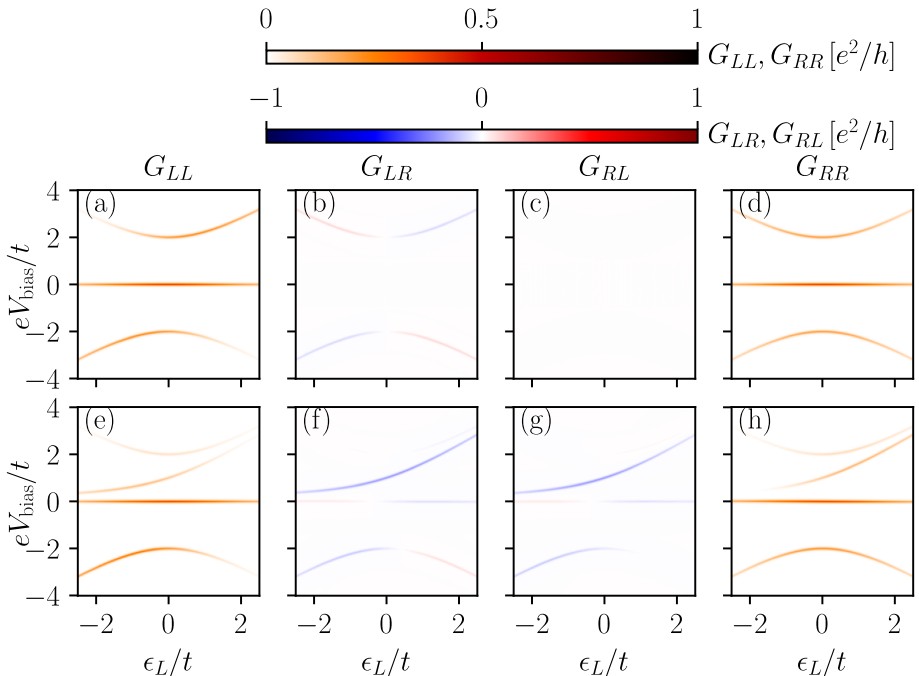

Figure 8: Comparison of the finite bias differential conductance between (top panels) a two-site spinless Kitaev chain and (bottom panels) a two-site spinful interacting chain while detuning the left on-site energy $\epsilon_L$. For transport simulations, we use dot-lead coupling $\Gamma = 0.0125t$ for both leads and temperature $T = 0.025t$ for both reservoirs.

is almost identical, we observe that the finite bias conductance spectroscopy can distinguish between two cases. Specifically, in Fig. 7(b) and (e), we show the local finite bias conductance spectroscopy for the spinless Kitaev chain and spinful interacting chain, respectively. Detuning both on-site energies, we observe that the local conductance for the spinful interacting chain, as shown in Fig. 7(e), features an additional trace of enhanced conductance at finite energy that moves down with decreasing $\epsilon$. The high charging energy of the dots prevents double occupation, allowing the ground state to only connect with triplet states by adding a single particle. This restriction on the transport process via triplet states explains the conductance asymmetry observed in Fig. 7(e) for the two-site spinful interacting chain with respect to bias voltage. The additional feature arises from the triplet states of the spinful interacting chain and is absent in the local conductance spectroscopy of the two-site spinless Kitaev chain. Finally, in Fig. 7(c) and (f), we examine the nonlocal differential conductance spectroscopy of both systems and observe that, similar to the local conductance signal, the transport processes via the triplet states in the spinful interacting chain can help distinguish between the two cases.

For completeness, we present all of the conductance matrix elements as we detune the left on-site energy $\epsilon_L$ in Fig. 8. Similar to Fig. 7(b,e), the local conductance elements $G_{LL}$ and $G_{RR}$ of the spinful interacting chain, shown in Fig. 8(e) and (h), feature an additional enhanced conductance trace compared to the spinless Kitaev chain case, shown in Fig. 8(a) and (d). This difference between two cases is also observed for the nonlocal conductance $G_{LR}$, as shown in Fig. 8(b) and (f). On the other hand, the most striking difference between the spinless Kitaev chain and the spinful interacting chain is observed in the nonlocal conductance $G_{RL}$. While the conductance vanishes entirely for the spinless Kitaev chain, as illustrated in Fig. 8(c), it remains finite for the spinful interacting chain. We observe that the transport via triplet states remains visible for the spinful interacting chain.

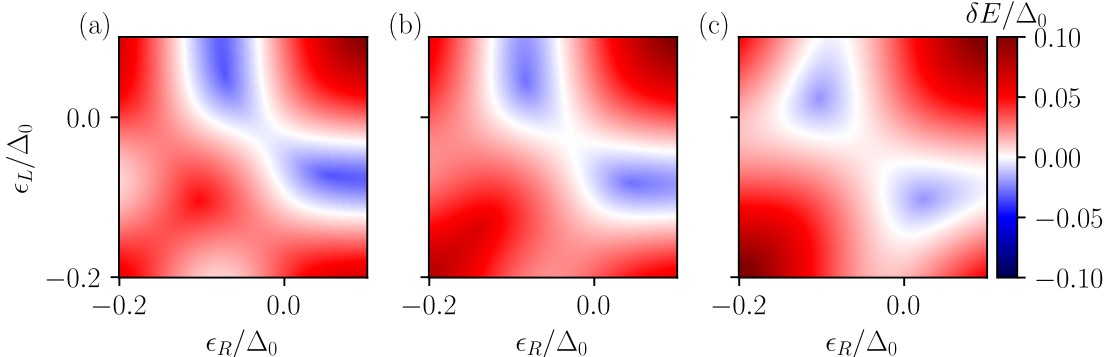

Figure 9: The change in connectivity of the charge stability diagram as the energy of the ABS is varied. The ground state switches its fermion-parity from (a) odd, with $\epsilon_M = \epsilon_M^* - 0.5\Delta_0$, to (c) even, with $\epsilon_M = \epsilon_M^* + 0.5\Delta_0$. This ensures that the degeneracy lines cross for a specific value of middle dot on-site energy $\epsilon_M^*$, as shown in panel (b). Here, we have $t_0 = 0.25\Delta_0$, $U = 0.1\Delta_0$, $\epsilon_M^* \approx -0.677\Delta_0$.

# B  The model including the Andreev bound state

In this appendix, we show the results obtained with the full model, including the proximitized quantum dot hosts an ABS. Here, ABS in the middle region mediates CAR and ECT between left and right quantum dots. The Hamiltonian for this system is given as [15, 39]

$$H = H_D + H_S + H_T \,, \tag{B.1a}$$

$$H_D = \sum_{\sigma, i=L,R} \epsilon_i n_{i\sigma} + \sum_{i=L,R} U_i n_{i\uparrow} n_{i\downarrow} \,, \tag{B.1b}$$

$$H_S = \epsilon_M \sum_\sigma n_{M\sigma} + \Delta_0 (c_{M\uparrow} c_{M\downarrow} + c_{M\downarrow}^\dagger c_{M\uparrow}^\dagger) \,, \tag{B.1c}$$

$$H_T = \sum_\sigma (t_0 c_{M\sigma}^\dagger c_{L\sigma} + t_0 c_{R\sigma}^\dagger c_{M\sigma}) + \text{H.c.} \,, \tag{B.1d}$$

where $H_D$ is the Hamiltonian of the quantum dots, $n_{i\sigma} = c_{i\sigma}^\dagger c_{i\sigma}$ is the spin-resolved electron occupation number on dot $i$, $U_i$ is the charging energy, $\epsilon_i$ is the on-site energy. $H_T$ describes the tunnel coupling between the outer dots and ABS in the middle, which features a spin-conserving hopping process with strength $t_0$. $H_S$ describes the middle dot that hosts an ABS in the low-energy approximation with an induced gap $\Delta_0$.

In Fig. 9(a-c), we demonstrate the evolution of the charge stability diagram while changing the energy of the ABS by varying $\epsilon_M$. Changing the ABS energy alters the effective parameters we use in the main text superconducting pairing $\Delta$ and normal hopping $t$. Similarly to the charge stability diagram of the effective model portrayed in Fig. 1, the connectivity of the charge stability diagram transitions from an odd ground state, as depicted in Fig. 9(a), to an even ground state, as illustrated in Fig. 9(c). Consequently, this ensures that a sweet spot condition can be achieved for any given $\epsilon_M$ value, as demonstrated in Fig. 9(b).

Furthermore, in Fig. 10, we depict the evolution of the charge stability diagram and the corresponding local differential conductance at the sweet spot as a function of the charging energy $U$ on the left and right quantum dots. Despite the increased complexity of the full model, qualitative features of the charge stability diagrams and corresponding local conductances exhibit similar behaviors to the effective model results, presented in the main text Fig. 2. As in the effective model, increasing the Coulomb interaction $U$ in the quantum dots

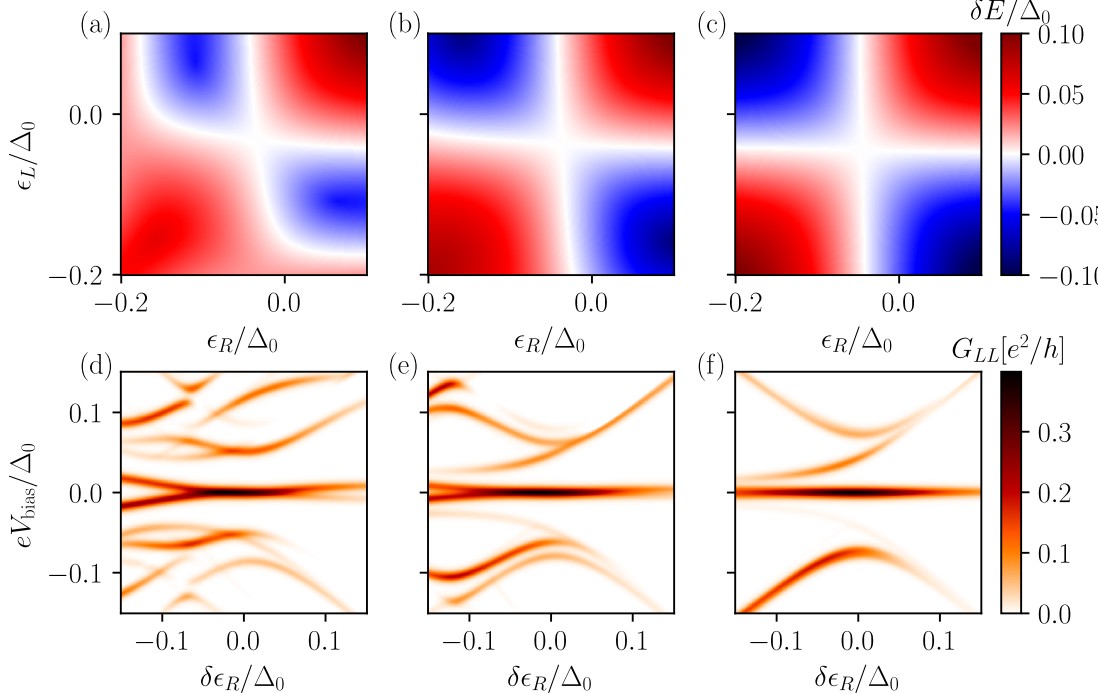

Figure 10: The evolution of charge stability diagram and local finite bias conductance $G_{LL}$ at the sweet spot for various charging energies for two-site chain with an ABS in the middle dot. For (a) and (d), we set $U = 0.15\Delta_0$; for (b) and (e), $U = 0.25\Delta_0$; and for (c) and (f), $U = 0.45\Delta_0$. Here, we use $t_0 = 0.25$, $T = 0.0025\Delta_0$, and $\Gamma = 0.00125\Delta_0$.

results in a widening separation between the regions of double occupancy and empty dots in the charge stability diagrams. Consequently, the degeneracy lines of the sweet spot crossing become straighter with increasing $U$, indicating the increased protection of the ground state degeneracy against local potential changes.

## C  Energy levels and many-body eigenstates of two-site spinful interacting chain

In this appendix, we list the eigenstate and eigenvalues of the spinful interacting chain in the limit of $U \to \infty$. At the sweet spot, i.e. $t = \sqrt{2}\Delta$ and $\epsilon_{L,R} = 0$, the spectrum exhibits three triply-degenerate manifolds. The many-body eigenstates for the ground state are already given in Eq. (3). Here, we show the eigenstates of the excited state manifolds. We start with $n = 1$, namely the triplet manifold

$$|n = 1, \downarrow\rangle = -|\downarrow\downarrow\rangle\,, \tag{C.1a}$$

$$|n = 1, \downarrow\rangle = +|\uparrow\uparrow\rangle\,, \tag{C.1b}$$

$$|n = 1, 0\rangle = -\frac{1}{\sqrt{2}}\left(|\downarrow\uparrow\rangle + |\uparrow\downarrow\rangle\right)\,. \tag{C.1c}$$

Finally, the eigenstates of the second excited state manifold, which consists of the bonding version of eigenstates of the ground state manifold

$$|n = 2, \downarrow\rangle = -\frac{1}{\sqrt{2}} \left(|\downarrow 0\rangle + |0\downarrow\rangle\right),\tag{C.2a}$$

$$|n = 2, \uparrow\rangle = -\frac{1}{\sqrt{2}} \left(|\uparrow 0\rangle + |0\uparrow\rangle\right),\tag{C.2b}$$

$$|n = 2, S\rangle = -\frac{1}{\sqrt{2}}|00\rangle - \left(\frac{1}{2}|\uparrow \downarrow\rangle - \frac{1}{2}|\downarrow\uparrow\rangle\right).\tag{C.2c}$$

In the subsequent appendices, we will make use of these many-body eigenstates to construct operators.

# D  Majorana Kramers-pair operators

In this appendix, we describe how we obtain the Majorana Kramers-pair operators. As described in the main text, the first excited state manifold is entirely composed of even fermion parity states. Consequently, our analysis of Majorana Kramers-pair operators excludes this manifold. We begin by constructing Majorana operators from the eigenstates of the many-body Hamiltonian. For a given spin projection, we define

$$\gamma_{1\sigma} = \sum_{n=0,2} e^{i\phi_{1,n}}|n, \sigma\rangle\langle n, S| + \text{H.c.},\tag{D.1a}$$

$$\gamma_{2\sigma} = \sum_{n=0,2} i e^{i\phi_{2,n}}|n, \sigma\rangle\langle n, S| + \text{H.c.},\tag{D.1b}$$

where $|n, S\rangle$ denotes the even parity state (singlet) and $|n, \sigma\rangle$ denotes the odd parity state with spin $\sigma$ in the $n^{\text{th}}$ manifold, and $\phi_n$ is an arbitrary phase. We find that for phase configuration

$$\phi_{1,n} = 0,\tag{D.2a}$$

$$\phi_{2,n} = \begin{cases} 0 & \text{if } n = 0, \\ \pi & \text{if } n = 2, \end{cases}\tag{D.2b}$$

$\gamma_{1\sigma}$ commutes with the number operator on the right site $n_R$, while $\gamma_{2\sigma}$ commutes with the number operator on the left site $n_L$. Consequently, we relabel $\gamma_{1\sigma}$ and $\gamma_{2\sigma}$ as $\gamma_{L\sigma}$ and $\gamma_{R\sigma}$, respectively. Eq. (5) of the main text, we present the decomposition of these operators in terms of constrained fermion operators.

# E  $\mathbb{Z}_3$ parity and parafermion operators

The three-fold degenerate structure of the many-body eigenstates of the two site chain presented in Sec. 3 and also quantum dot test presented in Sec. 4.1 signals a conserved symmetry of the system. As discussed in the main text, this symmetry is the generalized parity $P_{\mathbb{Z}_3} = e^{i\frac{2\pi}{3}\sum_j n_{j\uparrow} + 2n_{j\downarrow}}$, where $n_{j\sigma} = c_{j\sigma}^\dagger c_{j\sigma}$ is the spin-resolved number operator defined on dot $j$. Given the three-fold degenerate manifolds of the system under consideration, we express the parafermion operators as

$$\chi = \sum_{n=0}^{1} \sum_{p=-1}^{1} a_{n,p}|n, p\rangle\langle n, p+1 \bmod 3|,\tag{E.1}$$

where $n$ denotes the three-fold degenerate manifolds and $p$ represents the generalized parity eigenvalues of the states

$$P_{\mathbb{Z}_3} |n,p\rangle = \omega^p |n,p\rangle \,. \tag{E.2}$$

In Eq. (E.1), the coefficients $a_{n,p}$ are complex and satisfy $\prod_p a_{n,p} = 1$ for all $n$, ensuring that $\chi^3 = \mathbb{1}$.

Based on Eq. (E.2), we relabel the eigenstates according to their parity eigenvalues $P_{\mathbb{Z}_3}$. In the ground state manifold, given in Eq. (3), and the second excited state manifold, given in Eq. (C.2), we assign $\uparrow \mapsto p = 1$, $\downarrow \mapsto p = -1$ and $S \mapsto p = 0$. Furthermore, in the first excited state manifold given in Eq. (C.1), the labeling slightly varies due to the state $|n = 1, 0\rangle$: we assign $\uparrow \mapsto p = 1$, $\downarrow \mapsto p = -1$ and $0 \mapsto p = 0$.

Our procedure to construct parafermion operators relies on the following steps:

- Obtain the entire many-body spectrum and its eigenstates by exact diagonalization of the Hamiltonian.

- Label the eigenstates based on their energy-manifold $n$ and generalized parity eigenstate $p$.

- Construct the numerical matrices given in Eq. (E.1) using the eigenstates.

- Find the complex coefficients $a_{n,p}$ such that the corresponding parafermion operators commutes with a corresponding number operator.

- Express the resulting numerical matrix in terms of fermionic creation and annihilation operators.

In the next two subsections, we follow this procedure and construct the parafermion operators.

### E.1 Parafermion operators for two-site chain

For two-site chain, we have three three-fold degenerate manifolds, as shown in Fig. 3 in the main text. Following the method described above, we determine two sets of coefficients, $a_{n,p}$, which yield two parafermion operators. Each parafermion operator either commutes with the left or right number operator. The coefficients for the operator commuting with $\bar{n}_i$, which we name $a_{n,p}^i$, is

$$a_{n,p}^L = \begin{cases} -1 & \text{if } n = 0 \text{ and } p \neq -1\,, \\ -1 & \text{if } n = 2 \text{ and } p \neq 0\,, \\ 1 & \text{else}\,, \end{cases} \tag{E.3}$$

$$a_{n,p}^R = 1\,. \tag{E.4}$$

We note that the relative minus sign in $a_{n,p}^L$ is crucial for the commutation with the left number operator $\bar{n}_L$. Plugging the coefficients given in Eq. (E.3) into Eq. (E.1), we obtain the parafermion operators defined in Eq. (9).

## E.2 Different gauge choice for parafermion operators and parastatistics

We now use a different gauge choice for $\chi_L$ parafermion operator of the two-site spinful interacting chain, while keeping $\chi_R$ as the same defined in Eq. (E). This amounts to changing the coefficients $a_{n,p}^L$ defined in Eq. (E.3) as follows

$$a_{n,p}^L = \begin{cases} -\omega^{p+1} & \text{if } n = 2 \text{ and } p \neq 0\,, \\ \omega^{p+1} & \text{else.} \end{cases} \tag{E.5}$$

We express the resulting parafermion operator $\chi_L'$ in terms of constrained fermion operators as

$$
\begin{aligned}
\chi_L' &= \left(-\omega(1-\bar{n}_{R\uparrow}) + \left(\frac{1}{\sqrt{2}} + \omega\right)\bar{n}_{R\downarrow}\right)\bar{c}_{L\uparrow} - \left(\frac{\omega}{\sqrt{2}}\bar{c}_{L\uparrow}^\dagger + \frac{1}{\sqrt{2}}\bar{c}_{L\downarrow} + \omega^2 \bar{c}_{L\downarrow}^\dagger \bar{c}_{L\uparrow}\right)\bar{c}_{R\downarrow}^\dagger \bar{c}_{R\uparrow} \\
&\quad + \left(\frac{1}{\sqrt{2}} + \omega^2\right)\left(\bar{n}_{R\uparrow}\bar{c}_{L\uparrow}^\dagger \bar{c}_{L\downarrow} + \bar{n}_{L\uparrow}\bar{c}_{R\uparrow}^\dagger \bar{c}_{R\downarrow}\right) + \left(\frac{\omega^2}{2+\sqrt{2}} - \frac{1}{\sqrt{2}}\right)\left(\bar{n}_{R\downarrow}\bar{c}_{L\uparrow}^\dagger \bar{c}_{L\downarrow} + \bar{n}_{L\downarrow}\bar{c}_{R\uparrow}^\dagger \bar{c}_{R\downarrow}\right) \\
&\quad + \left(\left(\frac{\omega}{\sqrt{2}} - 1\right)\bar{n}_{R\uparrow} + (1-\bar{n}_{R\downarrow})\right)\bar{c}_{L\downarrow}^\dagger - \omega^2\left(\bar{c}_{L\uparrow}^\dagger \bar{c}_{L\downarrow} + \bar{c}_{R\uparrow}^\dagger \bar{c}_{R\downarrow}\right)\,.
\end{aligned}
\tag{E.6}
$$

We now explore the low-energy physics of the three-fold degenerate ground state manifold. To that end, we project the parafermion operators, specifically $\chi_R$ given in Eq. (9) and $\chi_L'$ given in Eq. (E.6), to the ground state manifold. These projected operators, denoted as $\tilde{\chi}_R$ and $\tilde{\chi}_L'$, still commute with their respective number operators. Additionally, we observe that the projected parafermion operators satisfy $\mathbb{Z}_3$ parastatistics

$$\tilde{\chi}_R \tilde{\chi}_L' = \omega \tilde{\chi}_L' \tilde{\chi}_R\,. \tag{E.7}$$

Mapping the parafermion operators on to the ground state and establishing that they obey $\mathbb{Z}_3$-parafermionic statistics allow us to derive the low-energy effective Hamiltonian using parafermion operators

$$\tilde{H} = -\frac{(2t + \sqrt{2}\Delta)}{3}\mathbb{1} + \frac{(t - \sqrt{2}\Delta)}{3}\left(\tilde{\chi}_L'^\dagger \tilde{\chi}_R + \tilde{\chi}_R^\dagger \tilde{\chi}_L'\right)\,. \tag{E.8}$$

We realize that the form of the low-energy Hamiltonian is similar to the low-energy Hamiltonian of a two-site $\mathbb{Z}_3$-parafermion chain. Similar to Eq. (10), the first term serves to ensure that at the sweet spot $t = \sqrt{2}\Delta$, the parafermions are decoupled.

## E.3 Parafermion operators for the quantum dot test

For the quantum dot test, we introduce a third quantum dot, labeled as $D$, which is attached to the right quantum dot of the two-site spinful interacting chain. In the absence of coupling between the right site and quantum dot $D$, the system exhibits three nine-fold degenerate manifolds, with each manifold labeled by its $\mathbb{Z}_3$ eigenstates. As the fermion-parity is conserved, we further order every eigenstate in each degenerate manifold according to its fermion-parity eigenvalue.

To ensure that the coupling between the test quantum dot and the two-site spinful interacting chain does not cause a splitting of the ground state degeneracy, we perform a unitary rotation on the ordered basis. This rotation is designed to ensure that the coupling, represented by $H_t = t_D \sum_\sigma \bar{c}_{R\sigma}^\dagger \bar{c}_{D\sigma} + \text{H.c.}$, maintains an identical matrix structure within each $\mathbb{Z}_3$ block. This property guarantees a three-fold degenerate structure in the entire spectrum for any value of $t_D$. Having established this basis, we proceed with the remaining steps of the procedure described above to determine the coefficients $a_{n,p}^i$. These coefficients ensure that the

resulting parafermion operators commute with either $\bar{n}_L$ or $\bar{n}_R$. Furthermore, we confirm that the obtained parafermion operators remain identical to their two-site version once the trace is taken over the test quantum dot. Details on the steps taken to get these coefficients and the expression of resulting parafermion operators in terms of fermionic creation and annihilation operators can be found in the code repository [38].

## F   Spin-orbit insensitivity of the degeneracies

This appendix demonstrates how a unitary transformation on the fermion operators in spin-space transforms the Hamiltonian given in Eq. (1) to one with spin-orbit interaction (cf. e.g. [40]). We follow the procedure outlined in Ref. [41] and perform a unitary transformation:

$$\begin{pmatrix} c_{L\uparrow} \\ c_{L,\downarrow} \end{pmatrix} = \begin{pmatrix} \tilde{c}_{L\uparrow} \\ \tilde{c}_{L,\downarrow} \end{pmatrix} \quad \text{and} \quad \begin{pmatrix} c_{R\uparrow} \\ c_{R\downarrow} \end{pmatrix} = \begin{pmatrix} \cos\left(\frac{\theta}{2}\right) & -\sin\left(\frac{\theta}{2}\right) \\ \sin\left(\frac{\theta}{2}\right) & \cos\left(\frac{\theta}{2}\right) \end{pmatrix} \begin{pmatrix} \tilde{c}_{R\uparrow} \\ \tilde{c}_{R\downarrow} \end{pmatrix}, \tag{F.1}$$

where $\theta$ is the spin-orbit angle relative to the basis of choice. Plugging the above identities into Eq. (1) we find the on-site term stays invariant

$$\sum_i \epsilon_i (c_{i\uparrow}^\dagger c_{i\uparrow} + c_{i\downarrow}^\dagger c_{i\downarrow}) = \sum_i \epsilon_i (\tilde{c}_{i\uparrow}^\dagger \tilde{c}_{i\uparrow} + \tilde{c}_{i\downarrow}^\dagger \tilde{c}_{i\downarrow}), \tag{F.2}$$

for the ECT term we have

$$t\left(c_{L\uparrow}^\dagger c_{R\uparrow} + c_{L\downarrow}^\dagger c_{R\downarrow} + h.c.\right) = t\cos\left(\frac{\theta}{2}\right)(\tilde{c}_{L\uparrow}^\dagger \tilde{c}_{R\uparrow} + \tilde{c}_{L\downarrow}^\dagger \tilde{c}_{R\downarrow} + h.c.)$$
$$+ t\sin\left(\frac{\theta}{2}\right)(-\tilde{c}_{L\uparrow}^\dagger \tilde{c}_{R\downarrow} + \tilde{c}_{L\downarrow}^\dagger \tilde{c}_{R\uparrow} + h.c.), \tag{F.3}$$

and finally for the CAR term

$$\Delta\left(c_{L\uparrow}^\dagger c_{R\downarrow}^\dagger - c_{L\downarrow}^\dagger c_{R\uparrow}^\dagger + h.c.\right) = \Delta\cos\left(\frac{\theta}{2}\right)(\tilde{c}_{L\uparrow}^\dagger \tilde{c}_{R\downarrow}^\dagger - \tilde{c}_{L\downarrow}^\dagger \tilde{c}_{R\uparrow}^\dagger + h.c.)$$
$$+ \Delta\sin\left(\frac{\theta}{2}\right)(\tilde{c}_{L\uparrow}^\dagger \tilde{c}_{R\uparrow}^\dagger + \tilde{c}_{L\downarrow}^\dagger \tilde{c}_{R\downarrow}^\dagger + h.c.). \tag{F.4}$$

The Coulomb term keeps its form only replacing $c_{i\sigma} \rightarrow \tilde{c}_{i\sigma}$. Collecting all terms we find the two-site Hamiltonian with spin-orbit hopping between the dots (cf. [40]).

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
