# Peer review of "Interaction-induced strong zero modes in short quantum dot chains with time-reversal symmetry"

_SciPost Physics, doi:SciPost Phys. 18, 206 (2025)_

## Round 1 · Referee Report · Anonymous (Referee 1) · 2024-9-5

Strengths

  1. Identification of a potential platform to study the physics of strong zero-modes

  2. Explicit constructions of the many-body zero modes operators

Weaknesses

  1. It is unclear if the platform can reproduce the physics of zero modes in realistic experimental regimes

  2. It is unclear if the transport signatures can be an indicator of many-body zero modes

Report

The authors study the spectrum and transport signatures of a double quantum dot coupled by elastic cotunneling and cross-Andreev reflection processes. The two processes can be controlled via a third proximitized quantum dot.  Similar setups have been proposed previously in the literature as a way to engineer sweet spots with a degenerate ground state so that they effectively reproduce the physics of Poor Man's Majoranas (PPMs) --- see e.g. Ref. 14. In the present work, the authors focus on the case with no magnetic field, so that the system is time-reversal symmetric, and address the role of the interaction.  They find that, for generic on-site interaction, it is possible to obtain a sweet spot by tuning \Delta/t. The sweet spot is correlated with the presence of a conductance peak which persists for a wider range of single energy detuning in the limit of large interactions where the degeneracy is better protected against local energy fluctuations. Finally, in the infinite-interaction limit, they identify an exact degeneracy of the whole spectrum and construct explicitly the associated Majorana and Parafermion zero modes. They show that a straightforward extension to longer chains fails to reproduce the physics of extended Kitaev chains or parafermionic models.

The results are sound and generically correct (modulo a couple of questions outlined below). They are an interesting extension of an idea already proposed in the literature (see e.g. Ref. 14) which provides a platform with the potential to explore the physics of Majorana and parafermions braiding. In this perspective, I find the identification of strong zero modes at infinite U, the most significant result in the manuscript, which could indeed open a new path to explore.

If the authors can better clarify the potential of this setup, given its limitations to extend it to longer chains, and the non-negligible effect of finite U (see below), I think the paper can be suitable for Scipost Physics. I would, otherwise, recommend publication in Scipost Core.

There are a couple of points that I think the authors should address.

  • Role of U in the physics of strong zero mode. While for U \to \infty the authors can identify the spectrum degeneracy and explicitly construct the corresponding zero modes operators, it is not clear how finite U will affect this. Tee authors analyze the sweet spot and ground state degeneracy at finite U, but there is no equivalent analysis of how the overall spectral degeneracy is affected by a fine U. The authors should comment on how the lifting of the degeneracy affects the strong zero modes and their physics. Furthermore, while the conductance seems a valid signature of the degeneracy of the ground state, can the authors comment on whether transport properties can give signatures of the many-body zero-mode?

  • Conductance computation in the presence of interactions. The authors compute the conductance using a rate equation approach. I do not expect this approach to be valid for the whole range of parameters. For example, for t=\Delta=0 where the setup reduces to a single dot coupled to a lead, the approach would miss Kondo correlations at low temperatures. I would naively expect the rate equation approach to hold for T>>\Gamma for the setup, but the data in Fig. 2 are for T~\Gamma. Can the authors clarify/justify the range of validity of the conductance calculation and ensure it is consistent with the simulations in the figures?

  • Minor point. I would suggest clarifying the role of topology in the setup. For example, arguing that the intrinsic "topology" of the charge stability diagram is the basis of the analogy with the Kitaev chain zero energy modes might be misleading. I would suggest clarifying that the system has no zero modes from (symmetry-protected) topology, but it is expected to emulate the physics (braiding properties) of such systems with an analogous degree of protection.

Requested changes

  1. Clarify the validity of the platform to observe strong zero-mode physics in an experimentally relevant regime.

  2. Address the impact of finite U (see report above)

  3. Clarify the validity of the conductance calculations.

Recommendation

Ask for major revision

---

## Round 1 · Referee Report · Anonymous (Referee 2) · 2024-10-7

Strengths

The paper generalizes the minimal Kitaev chain model to a situation with spin and interactions, and most importantly looks at the time-reversal symmetric case. This is a an interesting study which points out that besides having Kramers degenerated "poor man's" Majorana, the triply degenerate point can be mapping to parafermions.

Weaknesses

The latter part is considered in the limit of infinite U, which of course is a limitation.

Report

One question that arises is how this model compares to true topological Kramers Majorana bound states. The topological case requires two chanels (wires) and a certain relation between the spin-orbit interaction in the two chanels. In the model considered here, there is no spin-orbit interaction. A natural question that arises is if including spin-orbit coupling in the present model changes the physics? Maybe the authors may want to comment on this?

Recommendation

Publish (meets expectations and criteria for this Journal)

---

## Round 2 · Referee Report · Anonymous (Referee 1) · 2025-5-1

Strengths

  1. Identification of a potential platform to study the physics of strong zero-modes

  2. Explicit constructions of the many-body zero-mode operators

Weaknesses

-The signatures of the zero modes by the combination of the conductance peak and the particle-hole symmetry breaking due to the interaction term might be challenging to observe in experiments.
Yet it is a plausible experimental signature

Report

With the modifications in the resubmitted version and the responses to the Referees' comments, the paper fully meets the criteria for acceptance.

Recommendation

Publish (easily meets expectations and criteria for this Journal; among top 50%)

---

## Round 2 · Author Response

We thank both referees for their comments and overall positive evaluation. We give our point by point response to address referee's comments and questions below.

We also provide the redlined manuscript with the changes highlighted at this URL: https://surfdrive.surf.nl/files/index.php/s/KWtV3oxrg9hmhMx

Ref. 2:

The paper generalizes the minimal Kitaev chain model to a situation with spin and interactions, and most importantly looks at the time-reversal symmetric case. This is a an interesting study which points out that besides having Kramers degenerated "poor man's" Majorana, the triply degenerate point can be mapping to parafermions.

We appreciate the referee's positive comment on our work.

The latter part is considered in the limit of infinite U, which of course is a limitation.

We thank the referee for their remark. We would like point out that we have studied the three-fold ground state denegeracy or the case of finite $U$ in this system, as can be seen in Figs. 1 and 2. In this case, only the ground state is degenerate and as a result, we have weak zero modes. As we show in our manuscript, only in the limit $U\rightarrow \infty$, the full many-body spectrum becomes three-fold degenerate and as a result, we have strong zero modes. To emphasize the role of finite $U$ on the many-body spectrum, we have added panel a) to Fig. 3 and expanded the discussion.

One question that arises is how this model compares to true topological Kramers Majorana bound states. The topological case requires two chanels (wires) and a certain relation between the spin-orbit interaction in the two chanels. In the model considered here, there is no spin-orbit interaction. A natural question that arises is if including spin-orbit coupling in the present model changes the physics? Maybe the authors may want to comment on this?

We thank the referee for their question. The Majorana-Kramers pairs that are present in our system are induced by the presence of interactions. On the contrary, for single-particle regime Hamiltonians, the presence of spin-orbit interaction is required, which we don't need. Furthermore, due to time-reversal symmetry and the fact that we consider a one-dimensional system, we can gauge away the spin-orbit interaction without affecting the overall many-body spectrum. We demonstrate this gauge transformation explicitly in Appendix F where, starting from our model, we arrive at the general model for two quantum dots coupled by CAR and ECT and featuring spin-orbit interaction.

Ref. 1:

  1. It is unclear if the platform can reproduce the physics of zero modes in realistic experimental regimes

Current experimental implementations (see e.g. Refs. 10-12 in the manuscript) feature on-site Coulomb interaction that are at least two orders of magnitude larger than any other energy scale in the system. As shown in Fig. 2, for any finite $U$ the system can have a three-fold degenerate ground state in the presence of time-reversal symmetry. The lack of degeneracy in the excited states would however not affect the zero-bias features seen in conductance. To understand this, consider e.g. PMMs in presence of magnetic fields. These also lack a strong degeneracy and are hence weak zero modes. Yet, they show zero-bias features. The lack of degeneracy in the excited spectrum is also visible in Figs. 2d) - f) of the manuscript, where one can recognize the excited states approaching each other in energy but not becomming strictly degenerate. To however expand on this aspect, we added Fig. 3 a) that shows the spectrum in dependence of $U$ directly.

  1. It is unclear if the transport signatures can be an indicator of many-body zero modes

We thank the referee for their remark. The distinct feature of many-body zero modes in quantum dot-based systems is the presence of Coulomb interaction, which results in particle-hole symmetry breaking. As demonstrated in our paper, finite bias transport can reveal whether particle-hole symmetry is broken. When this observation is combined with the presence of a robust zero bias peak in transport measurements, it can indeed serve as an indicator of a many-body zero mode.

Role of U in the physics of strong zero mode. While for U \to \infty the authors can identify the spectrum degeneracy and explicitly construct the corresponding zero modes operators, it is not clear how finite U will affect this. Tee authors analyze the sweet spot and ground state degeneracy at finite U, but there is no equivalent analysis of how the overall spectral degeneracy is affected by a fine U. The authors should comment on how the lifting of the degeneracy affects the strong zero modes and their physics.

We thank the referee for their comment. For any finite $U$, the many-body energy spectrum at the sweet spot is not entirely degenerate. As a result, the system features weak zero modes, i.e. only the ground state is degenerate and excited state manifolds will no longer be degenerate. However, it is still possible to construct relevant operators describing the weak zero modes by using the degenerate ground state eigenstates only. As a result, the ground state degeneracy will not be protected against local perturbations anymore, as we show in Figs. 2d) - f) of the manuscript. However, the protection of the ground state degeneracy and the degeneracy of the excited state manifold will be restored asymptotically as $U$ increases. To illustrate this aspect more clearly, we added Panel a) in Fig. 3 that shows the many-body spectrum as a function of $U$ directly.

Furthermore, while the conductance seems a valid signature of the degeneracy of the ground state, can the authors comment on whether transport properties can give signatures of the many-body zero-mode?

The many-body nature of the problem becomes apparent due to the lack of particle-hole symmetry in the transport simulations. Specifically, the finite bias conductance simulations shown in Figs. -- show the triplet conductance signature for $eV_{\textrm{bias}}>0$, but not for $eV_{\textrm{bias}}<0$. This lack of particle-hole symmetry is a strong signature of the many-body nature of the underlying system and it is accessible via transport measurements.

Conductance computation in the presence of interactions. The authors compute the conductance using a rate equation approach. I do not expect this approach to be valid for the whole range of parameters. For example, for t=\Delta=0 where the setup reduces to a single dot coupled to a lead, the approach would miss Kondo correlations at low temperatures. I would naively expect the rate equation approach to hold for T>>\Gamma for the setup, but the data in Fig. 2 are for T~\Gamma. Can the authors clarify/justify the range of validity of the conductance calculation and ensure it is consistent with the simulations in the figures?

We thank the referee for their comment. The referee is right pointing out that the rate equation approach is valid in the parameter regime $T\ll \Gamma$. We have used a lower value for the temperature to have sharper conductances features. Increasing the temperature will only change the conductance features qualitatively, i.e. broadening the conductance lines and/or reducing their magnitude. Following the referee's comment, we have changed the parameters of our conductance simulations such that $T > \Gamma$ throughout the manuscript. As visible, this change does not alter the main features of our results.

  • Minor point. I would suggest clarifying the role of topology in the setup. For example, arguing that the intrinsic "topology" of the charge stability diagram is the basis of the analogy with the Kitaev chain zero energy modes might be misleading. I would suggest clarifying that the system has no zero modes from (symmetry-protected) topology, but it is expected to emulate the physics (braiding properties) of such systems with an analogous degree of protection.

We thank the referee for pointing out the confusion our phrasing may lead to. To clarify, by "topology of the charge-stability diagram" we meant to describe the connectivity of the charge stability diagram and the change it undergoes when sweeping across the sweet spot. To avoid such confusion, we have removed the word "topology" from our description.

---

## Round 2 · List of Changes

To summarize our changes to the manuscript: * We have changed Fig. 3, adding panel a) to show the spectrum in dependence of $U$, and added an additional sentence in the caption of Fig. 2 meant to illustrate the changes in the finite $U$ case. * We have removed the word "topology" in our explanation on page 3. * We have changed the parameters of our conducatance simulations to be $T=2\Gamma=0.0125t$. This we clarify in the caption of Fig. 2. * We expand on why we can gauge away spin-orbit interaction in our system in the newly added Appendix F. * Added an ORCID for co-author

---

## Editorial Decision

published